

# Effect of *Citrus aurantium* juice as a disinfecting agent on quality and bacterial communities of striped catfish steaks stored at −20 °C

Kajonsak Dabsantai and Thitikorn Mahidsanan

Established Faculty of Innovative Agriculture and Technology, Institute of Interdisciplinary Studies, Rajamangala University of Technology Isan, Nakhon Ratchasima, Thailand

Corresponding author
Thitikorn Mahidsanan,
thitikorn_mahi@hotmail.com,
thitikorn.ma@rmuti.ac.th

## ABSTRACT

Sodium hypochlorite is generally used as a disinfectant in washing of freshwater fishes where the safety aspect of health is of concern. Although plant-based essential oils and synthetic chemical agents have been applied, they might contain toxic substances, are expensive and can cause undesirable quality. This research aims to fill the knowledge gap necessary to validate *Citrus aurantium* juice as a disinfecting agent for preserving striped catfish steaks at −20 °C for 28 days. Fifty (50) ppm sodium hypochlorite was used as a commercial disinfectant (control). The results showed that a negative color characteristic (higher a* and increased b*) was found in the control but not in striped catfish steaks immersed in *C. aurantium* juice (TM) on days 14 and 28. No significant differences were found in the peroxide value among the treatments on days 14 and 28 ($P > 0.05$). A lower accumulation of trichloroacetic acid soluble peptides was detected in TM but not in control, while total volatile basic nitrogen of all treatments was up to standard of fish quality during storage. Contrastingly, the total viable count of both treatments increased to >7.0 log CFU/g on day 28 which did not meet the edible limit of standard for freshwater fishes. The spoilage microbial community was observed on days 0 and 28 of storage which showed a decrease in relative abundance of *Acinetobacter, Pseudomonas, Brochothrix, Lactococcus, Carnobacterium, Psychrobacter*, and *Vagococcus* as found in TM on day 28, when compared to the control. Thus, these results implied that *C. aurantium* juice could replace sodium hypochlorite as an alternative disinfecting agent to control the microbiological spoilage and physico-chemical quality of striped catfish steaks.

## INTRODUCTION

*Pangasianodon hypophthalmus* or Striped catfish (named Pla Sawai in Thai), is a large freshwater catfish that can be found in Thailand, Vietnam, Malaysia, China, and Indonesia. Striped catfish usually prefer a tropical climate where the pH values of freshwater are approximately 6.5–7.5 and the temperatures about 22–26 °C (*Roberts &*

*Vidthayanon, 1991*; *Singh et al., 2011*). This fish is an economically significant aquaculture species because of its high yield percentage and value as a product in food industry (*Singh et al., 2011*). During processing, striped catfish can be converted from fresh whole fish to produce various convenience foods such as fish fillets, dresses, sticks, balls, and steaks (*FAO, 2006*; *Islami et al., 2014*; *Rathod et al., 2018*).

In general, freshwater fish are highly susceptible to spoilage from microbial growth and biochemical products due to their intrinsic (water content, pH, nutrient properties) and extrinsic factors (temperature and package conditions) which provide suitable conditions for the growth of spoilage bacteria (*Taormina, 2021*). *Thi et al. (2013)* evaluated the microbiota of catfish (*Pangasius hypophthalmus*) and revealed that spoilage bacteria including *Aeromonas*, *Acinetobacter*, *Lactococcus* and *Enterococcus* were prevalent at various processing steps on the processing lines. As previously reported, a range of chilling temperatures has been used to preserve parts of fish products, even though these conditions might negatively affect the sensory characteristics, microbiological and biochemical parameters, resulting in a short shelf life of products (*Binsi et al., 2014*; *Deepitha et al., 2021*; *Rao, Murthy & Prasad, 2013*). To extend the shelf life, *Tong Thi et al. (2016)* suggested that vacuum packaging and freezing could be used to control the microbiota of stored catfish.

A retardation of deterioration of fresh fish has been considered, especially chlorine-based sanitizers, which are implemented as disinfecting agent in frozen fish and fishery products (*World Health Organization, 2009*). The use of food additives produced from synthetic chemical agents, especially sodium hypochlorite which is commonly used for preserving fish products, must be limited based on the safety concerns of foods and health. A previous report on residual chlorine toxicity in aquatic systems and environment considered the need for close scrutiny of disinfection procedures in aquatic life (*Brungs, 1973*). In the same way, another study revealed that the use of chlorinated washing water had no effect on the spoilage microbiota in *Pangasius hypophthalmus* fillets (*Thi et al., 2013*). To overcome the hazard of chemical preservatives, increased research has emphasized the use of plant-based and/or natural preservative agents (*Li et al., 2020b*; *Zhuang et al., 2019*). Furthermore, lemon juice, tamarind pulp, lemon grass, and banana leaves have been used as washing agents for minimizing muddy taint associated with fish tissue (*Bakar & Hamzah, 1997*; *Mohsin, Bakar & Selamat, 1999*).

*Citrus aurantium*, known as sour orange or bitter orange, grows in semitropical and temperate areas. It can be found in parts of Thailand and has high commercial value. In addition, it has been revealed that *Citrus* flavonoids have wide spectrum of antimicrobial, antidiabetic, anticancer and antioxidant activities. Parts of *C. aurantium*, including flower, leaf, ripe and unripe peel essential oils have been recently investigated based on the chemical and antimicrobial activities in extending shelf-life, which contributes significantly to the quality of food products (*Azhdarzadeh & Hojjati, 2016*; *Değirmenci & Erkurt, 2020*; *Pratama, Premjet & Choopayak, 2019*; *Wen et al., 2021*). Previous studies have indicated that essential oils extracted from plants including oregano, sweet bay, thyme, garlic, clove, cumin, spearmint, and *Allium* species have been used as preservative agents for fishes stored under chilling temperature (*Attouchi & Sadok, 2012*;

*Cai et al., 2015*; *Erkan, 2012*; *Vatavali et al., 2013*; *Xu et al., 2015*). *Viji et al. (2015)* showed that ethanolic extract of *C. aurantium* peel enhanced the storage stability and extended the shelf life of mackerel (*Rastrelliger kanagurta*) by 2 days during storage at 0–2 °C. *Yerlikaya et al. (2015)* presented citrus peel extracts incorporated ice cubes use in controlling the biochemical indices in common pandora (*Pagellus erythrinus*). Besides, *Oncorhynchus mykiss* fillets coated with chitosan enriched with fenugreek essential oil and immersed in *C. aurantium* juice concentrate were investigated by *Tooryan & Azizkhani (2020)*. However, plant essential oils obtained by chemical solvent extraction have high cost, may have toxic substances, and can cause unstable product quality. Consumers are avoiding food raw materials treated with synthetic chemical preservatives and thus the natural choice implemented at the household level and industry is required (*Chemat, Vian & Cravotto, 2012*; *Płotka-Wasylka et al., 2017*). In an attempt to prevent the risk of short shelf-life and to reduce the cost of production, *C. aurantium* juice could be further applied as a new alternative for natural sanitizers owing to its antimicrobial activity (*Karabıyıklı, Değirmenci & Karapınar, 2014*).

To the best of our knowledge, there are no previous studies on the effect of *C. aurantium* juice as disinfecting agent on the quality of striped catfish during frozen storage. Besides, the potential effect of *C. aurantium* juice on the bacterial community in striped catfish has not been revealed yet in academic databases. Therefore, the aim of this study was to validate the *C. aurantium* juice as a preservation agent for striped catfish steaks at −20 °C. The changes in microbiological and physico-chemical quality in samples during a 28-day frozen storage were characterized. Additionally, the bacterial community was assessed by Illumina-MiSeq high throughput sequencing to evaluate the preservative effects of *C. aurantium* juice.

## MATERIALS AND METHODS

### Preparation of *C. aurantium* juice and fish samples

Fresh *C. aurantium* was obtained from a local agricultural farm in the Khamthaleso sub-district, Nakhon Ratchasima province, Thailand. It was rinsed with tap water and sliced. The juice extract was prepared by mechanical machine (Severin, Sundern, Germany). Subsequently, the juice was filtered by sterile gauze for further experiment.

Striped catfish steak was purchased from Sura-Nakhon processing line, Nakhon Ratchasima province, Thailand. The mean length, weight, and thickness of steaks were 10–13 cm, 80–105 g, and 0.5–1.0 cm, respectively. The steaks were washed twice with reverse osmosis water and dried on a cleaned tray for 10 min. The striped catfish steaks were divided into two groups: the first group (control) was immersed in a commercial sanitizer, 50 ppm sodium hypochlorite for 10 min, and the second group (TM) was immersed in a natural sanitizer, *C. aurantium* juice for 10 min. Each steak was packed in a vacuum sterile bag and stored at −20 °C. The samples of each group were randomly selected for physico-chemical quality analysis and microbial enumeration during storage on (0, 7, 14, 21, and 28 days). Bacterial communities were analyzed on days 0 and 28.

## Physico-chemical analysis
### Color measurement
The colors of each treatment were analyzed using a Chroma meter CR-410 (Konica Minolta, Toyko, Japan). The color parameters, including $L^*$ (brightness), $a^*$ (redness/greenness), and $b^*$ (yellowness/blueness), were observed and their mean values were then used to represent the color values.

### Texture analysis
The texture profiles were analyzed by using a texture analyzer (CT3 10K, BROOKFIELD, USA), equipped with a specific cylindrical probe (TA41). Each sample was then compressed under the following conditions: a pre-test speed of 2.0 mm/sec, a test speed of 1.0 mm/sec, a post-test speed of 2.0 mm/sec, a compression of 25%, and a trigger force of 5 g. The hardness, adhesiveness, springiness, and cohesiveness were also recorded.

### Determination of peroxide value (PV)
The PV was measured following the method described by *AOAC (2008)*. The sample (0.5 g) was mixed with 10 mL of glacial acetic acid-chloroform mixture (3:2, v/v). Saturated potassium iodide solution (0.5 mL) was then added and the mixture allowed to stand for 15 min in darkness. After that, 10 mL of distilled water was added, and the free iodine was titrated with 0.01 mol/L sodium thiosulfate solution with the addition of 1% (w/v) starch solution as an indicator. Results were also expressed as meq/g sample.

### Analysis of pH values and total volatile basic nitrogen (TVB-N)
One gram of each treatment was homogenized with 10 mL distilled water for 1 min. The mixture was then centrifuged at $4,000 \times g$ for 10 min, and the pH value of the supernatant was measured by pH meter (Fisher Scientific model AB15, Thermo Fisher Scientific, Waltham, MA, USA).

The TVB-N was measured following the method described by *Malle & Poumeyrol (1989)* with slight modifications. Briefly, 200 mL of 7.5% (w/v) trichloroacetic acid solution was added to 100 g of fish sample. After homogenization, the mixture was centrifuged at $400 \times g$ for 5 min and then filtered by Whatman No. 1 filter paper. Distillation was then performed using Kjeldahl apparatus (Vapodest 30 s; Gerhardt, Hessen, Germany). Ten milliliters of filtrate were loaded into distillation tube followed by 6 mL of 10% (w/v) sodium hydroxide. A beaker containing 10 mL of 4% (w/v) boric acid and 0.04 mL of methyl red and bromocresol green indicator was used under the condenser for titration of ammonia. Distillation was started and steam distillation further continued until a final volume of 50 mL was obtained in its beaker (40 mL of distillate). The boric acid solution turned green when alkalinized by the distilled TVB-N which was titrated with aqueous 0.01 mol/L hydrochloric acid solution. Complete neutralization was obtained when the color turned pink on the addition of a further drop of hydrochloric acid. Results were expressed as mg/100 g sample.

### Measurement of trichloroacetic acid (TCA)-soluble peptides

The measurement of TCA-soluble peptides was performed by the method of *Jia et al. (2019)* with slight modifications. Two grams of each sample were homogenized with 18 mL of cold 5% (w/v) TCA and stored in an ice bath for 30 min. The mixture was centrifuged at $10,000 \times g$ for 10 min at 4 °C. The TCA-soluble peptides were analyzed by Lowry method (*Lowry et al., 1951*). Samples were mixed with C solution (1 mL; mixture of 50 mL of A solution, 0.5 mL of $B_1$ solution and 0.5 mL of $B_2$ solution). D solution (0.1 mL) was then added after 10 min. The absorbance was recorded at 750 nm after 30 min against a blank sample. The TCA-soluble peptides were also expressed as µg tyrosine/g sample.

## Microbiological analysis

### Microbial enumeration

Briefly, 25 g of each sample was homogenized with 225 mL sterile saline to produce the first dilution, and 10-fold dilutions were then made. Total viable count (TVC) was determined by spreads on plate count agar incubated at 37 °C for 24 h. The results are expressed in log CFU/g.

### Bacterial community by Illumina-MiSeq high throughput sequencing

To analyze bacterial community, the control and TM kept for 0 and 28 days, respectively, were chosen. Metagenomic DNA of each treatment was isolated using DNeasy Blood & Tissue Kits (Qiagen, Hilden, Germany) according to the manufacturer's protocols. Briefly, 25 mg of pooled sample taken from six steaks of each treatment were randomly extracted and the quality of the extracted DNA was determined by DeNovix QFX Fluorometer. The prokaryotic 16S rRNA gene at V3-V4 region was performed using the Qiagen QIAseq 16S/ITS Region panel. The PCR program was as follows: 95 °C for 2 min, 12 cycles at 95 °C for 30 s, 50 °C for 30 s, 72 °C for 2 min and 72 °C for 7 min. The DNA amplicon was then purified with QIAseq beads at 1.1X volume to clean up contaminated PCR products. The 16S rRNA amplicons were labeled with different sequencing adaptors using QIAseq 16S/ITS Region Panel Sample Index PCR Reaction, with the PCR program as follows: 95 °C for 2 min, 19 cycles at 95 °C for 30 s, 60 °C for 30 s, 72 °C for 2 min and 72 °C for 7 min. The DNA libraries were purified with one round of QIAseq beads at 0.9X volume and eluted in 25 µl of nuclease-free water. The quality and quantity of approximately 630-bp of DNA libraries were evaluated using QIAxcel Advanced and DeNovix QFX Fluorometer, respectively. The 16S rRNA libraries were sequenced using an Illumina Miseq 600 platform (Illumina, San Diego, CA, USA).

Bioinformatics analysis was done while the raw sequences were categorized into groups based on the 5′ barcode sequences. The sequences were processed following DADA2 v1.16.0 pipeline (https://benjjneb.github.io/dada2/). The DADA2 pipeline describes microbial diversity and community structures using unique amplicon sequence variants (ASVs). Microbial taxa were classified from Silva version 138 as a reference database.

## Statistical analysis

The experiment was conducted in triplicate. Differences in means of physico-chemical parameters and microbial enumeration were subjected to analysis of variance (ANOVA) followed by Duncan's multiple range test (DMRT), using IBM SPSS statistics 23. A probability level of $P < 0.05$ was considered statistically significant. In terms of bacterial community, principle coordinate analysis (PCoA) plots including unweighted UniFrac, weighted UniFrac, GUniFrac with alpha 0.5, and Bray-Curtis distance were used to calculate distances between samples based on the similarity of their members.

# RESULTS AND DISCUSSIONS

## Color characteristics

The parameters of color, such as L* (lightness), a* (redness-greenness), and b* (yellowness-blueness) of striped catfish steak during storage at −20 °C were evaluated and the results shown in Fig. 1. The values of a* and b* of the control treatment were significantly higher ($P < 0.05$) than those of TM on days 14 and 28. In the case of day 21, a decrease in a* and b* was shown in both treatments when compared to day 14 ($P < 0.05$). The oscillatory pattern was found in all color parameters. The differences observed in color parameters in this study were related to the storage times in that they became redder (higher a*) and more yellow (increased b*). This phenomenon might have been due to the lipid oxidation, which represents a commercial disadvantage during storage at cold conditions, in agreement with previous study in fish (*Cakli et al., 2006*). Similar characteristics were found in this work and that of *Álvarez et al. (2008)* who evaluated the L* and a* on the ventral side of *Sparus aurata* during ice storage and reported that those color parameters displayed a very significant negative correlation with storage time resulting in the discoloration of gilthead seabream skin. Likewise, *Sáez et al. (2015)* studied the effects of vacuum and modified atmosphere on color changes during cold storage of *Argyrosomus regius* fillets and found the values of a* were consistently negative, indicating lipid oxidation. Regarding the disadvantage of chlorine treated fish characteristic, *Chuesiang, Sanguandeekul & Siripatrawan (2020)* found that an increase in the b* value of sodium hypochlorite-treated Asian seabass (*Lates calcarifer*) fillets was observed during storage because of the interference from the sodium hypochlorite solution. *Naha, Varalakshmi & Velmathi (2019)* clarified that the sodium hypochlorite-bleaching ability may interfere with the visible light absorption of the fish samples due to breaking of the chemical bonds of the colored compounds contained in the samples. Consequently, this negative characteristic may affect the consumer buying decision. The discussion of lipid oxidation will be presented based on the peroxide values in a later section.

## Texture profiles

Texture parameters such as hardness, adhesiveness, springiness, and cohesiveness of the striped catfish steak were monitored and results revealed in Fig. 2. The control treatment was observed to have a higher adhesiveness and springiness on day 28 ($P < 0.05$) when compared to day 0. However, the hardness of TM was lower than that of the control treatment on day 28 ($P < 0.05$). This might be attributed to the alteration of protein

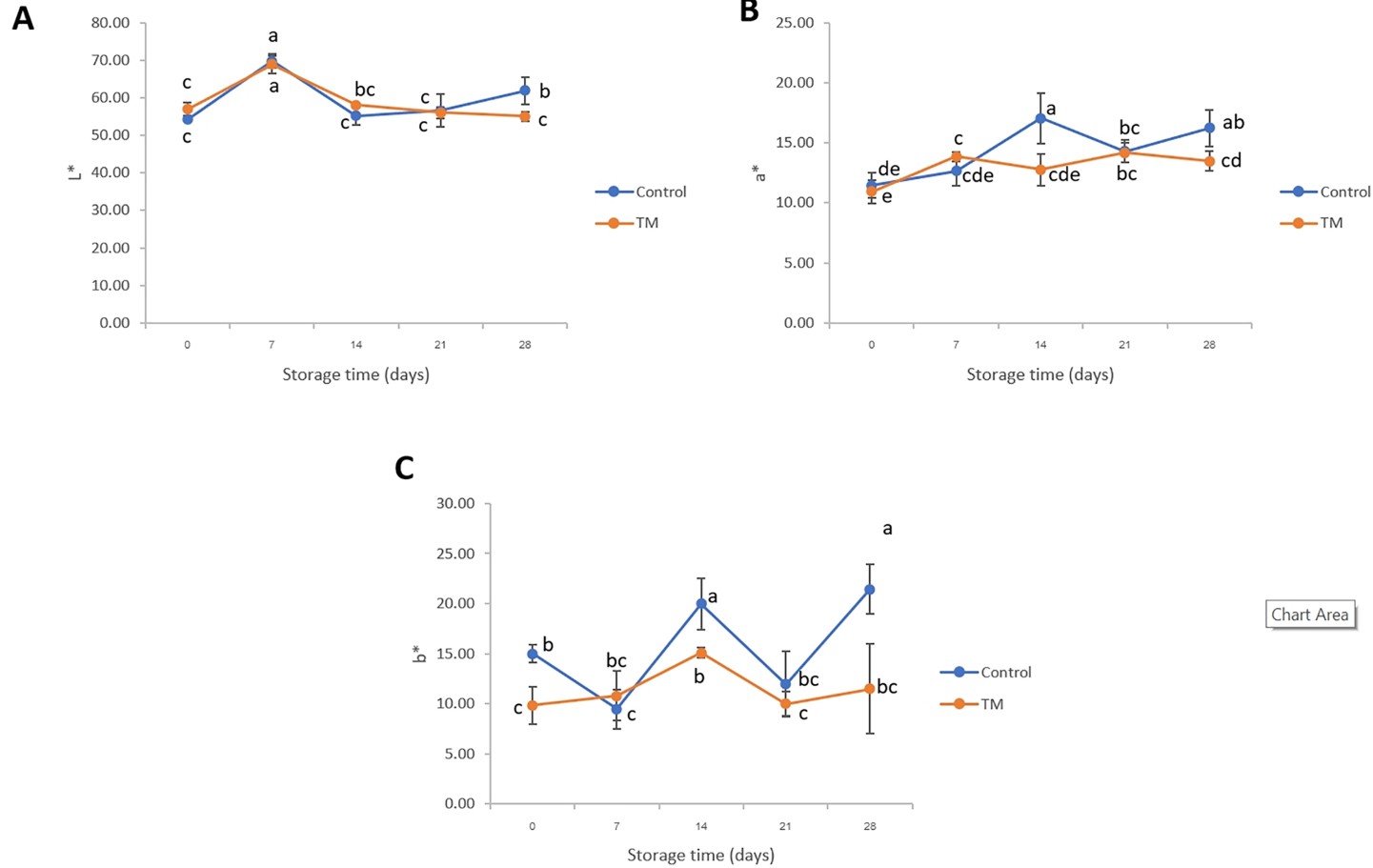

**Figure 1 Changes in L* (A), a* (B), and b* (C) of the control and *C. aurantium* juice-immersed striped catfish steaks during storage at −20 °C for 28 days.** Results are presented as mean ± standard deviation. Means in each parameter followed by different lowercase letters are significantly different (*P* < 0.05) according to DMRT.

structure. The change of protein oxidation and denaturation (disulfide (SS), carbonyl contents, salt-soluble protein (SSP) and $Ca^{2+}$-ATPase activity) might have occurred during storage at −20 °C (*Lu et al., 2021*). However, on days 0–21, all samples had no significant difference (*P* > 0.05) in all textural parameters implying that *C. aurantium* juice immersion did not significantly change the textural parameters (*P* > 0.05) during storage for 21 days as compared to sodium hypochlorite immersion. Despite the sodium hypochlorite not affecting the texture characteristics of striped catfish steak during storage, the safety aspect of antimicrobial agents has been of concern in the European Union due to lingering toxic residues at the food consumption stage. In fact, the use of sodium hypochlorite in the presence of organic matter promotes trihalomethane formation by an oxidation reaction, and it has been extensively discussed as a great disadvantage, especially due to carcinogenic properties (*EFSA, 2006*; *World Health Organization, 2009*). In the same manner, *Hernández-Pimentel et al. (2020)* who aimed to reduce the use of sodium hypochlorite revealed that neutral electrolyzed water could be applied as an alternative antimicrobial

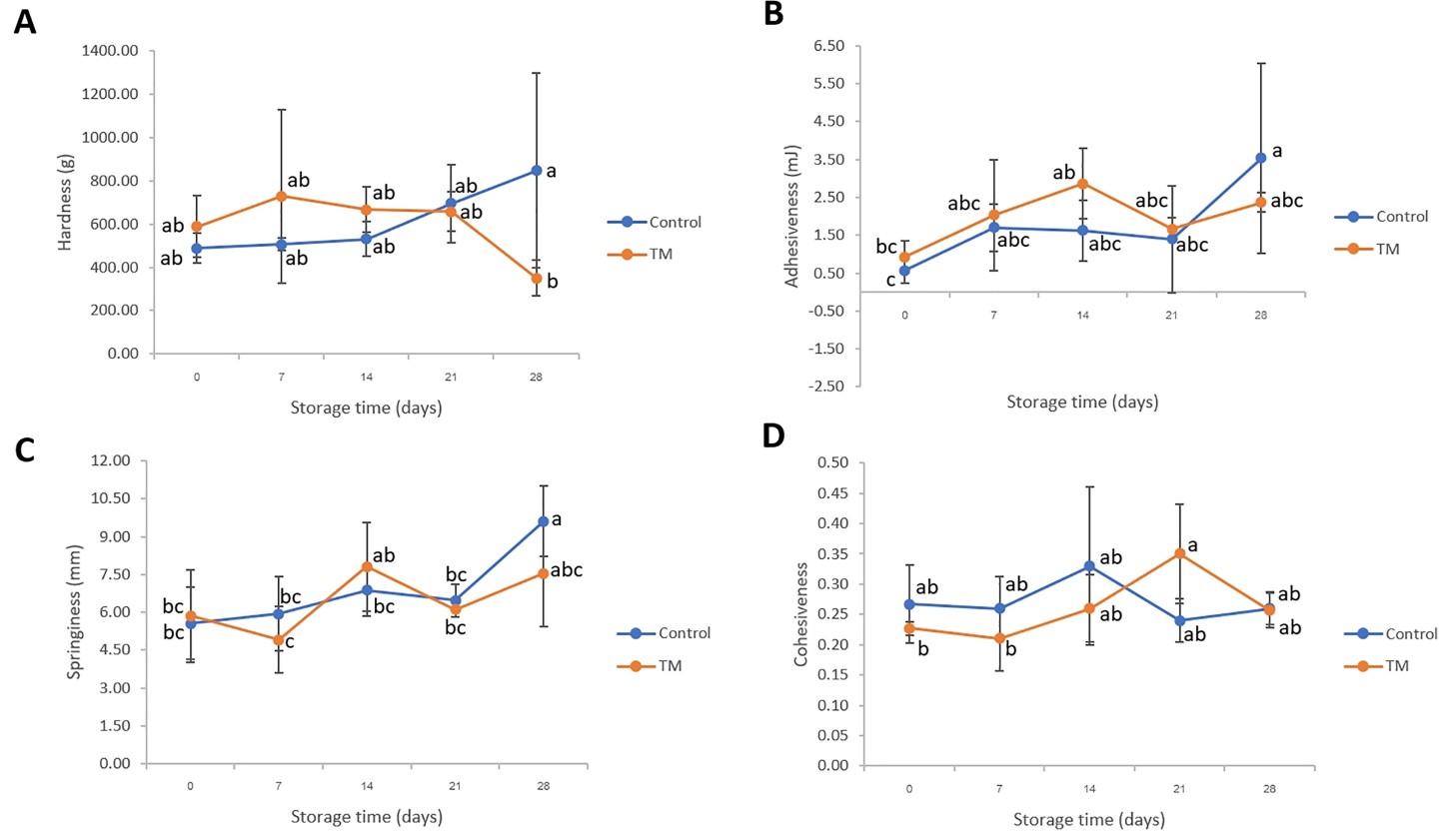

**Figure 2 Changes in hardness (A), adhesiveness (B), springiness (C), and cohesiveness (D) of the control and *C. aurantium* juice-immersed striped catfish steaks during storage at −20 °C for 28 days.** Results are presented as mean ± standard deviation. Means in each parameter followed by different lowercase letters are significantly different (*P* < 0.05) according to DMRT.

agent during chicken meat processing because of safe handling, high availability, low toxicity, low corrosion capacity, and no by-product generation.

## pH, peroxide values, TVB-N, and TCA-soluble peptides

The change in pH value of each treatment is illustrated in Fig. 3A. On day 0, the pH of the control and TM was 6.68 ± 0.01 and 5.71 ± 0.25, respectively. This result indicated that the pH value of TM was lower than that of the control (*P* < 0.05) due to the low pH condition of *C. aurantium* juice. No significant differences (*P* > 0.05) were then indicated in the pH values between the control treatment and TM during storage on days 14 and 21. After that, a decrease in pH was found in both treatments at the end of storage (day 28) due to the ATP decomposition, lactic acid, glycolysis, and pyrophosphate accumulation in fish muscle during storage (*Li et al., 2020b*). Similarly, the report of *Li et al. (2022)* showed that the pH value of *Micropterus salmoides* decreased slightly up to the 14 days (pH < 7.00) during frozen storage at −30 °C which might have been caused by microbial fermentation of carbohydrates resulting to organic acids production in fish muscle (*Khan, Parrish & Shahidi, 2005*). However, the variations in the pH value between these investigations are probably owing to the differences in the geographical location, catching season, water composition, and fish size (*Malik et al., 2021*).

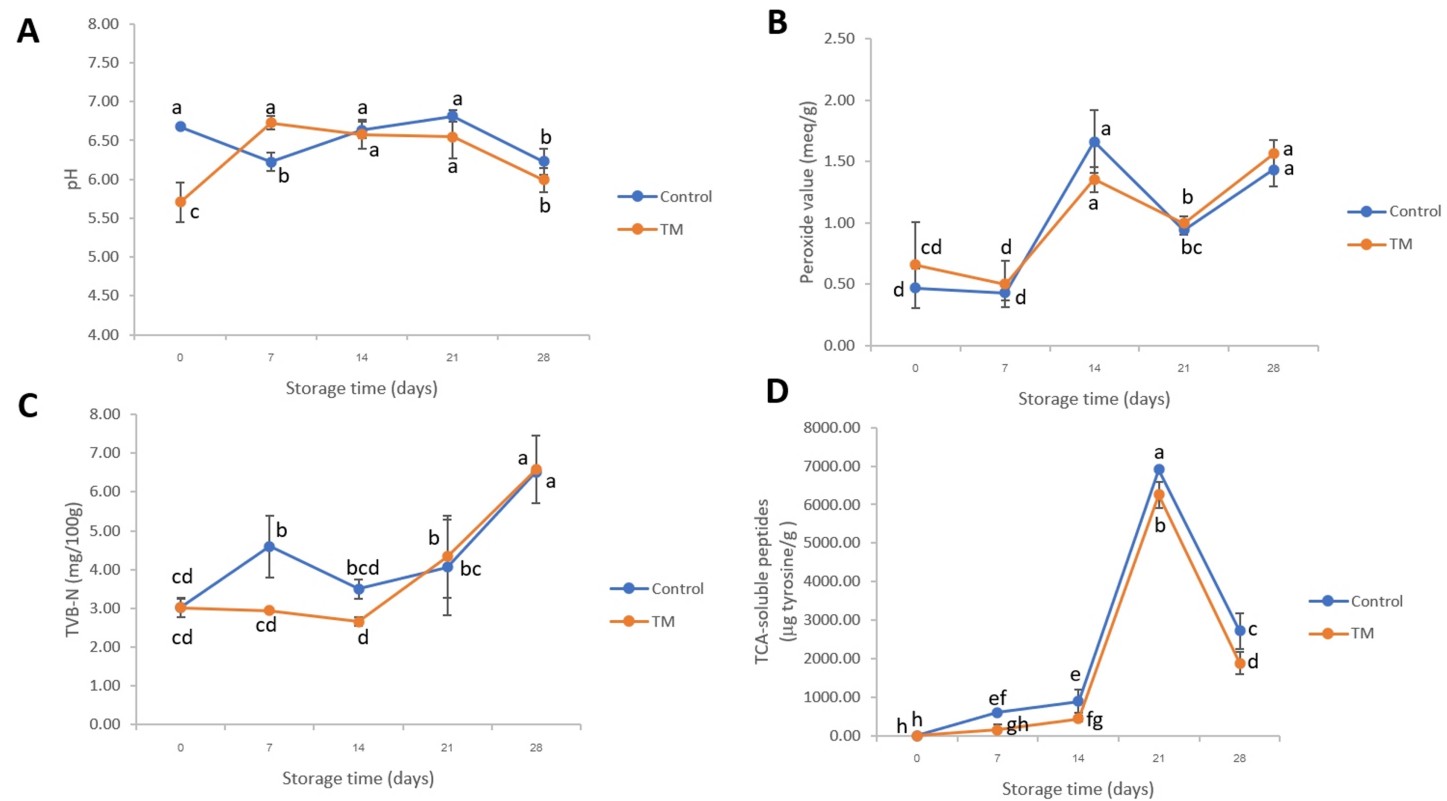

**Figure 3 Changes in pH values (A), peroxide values (B), TVB-N values (C), and TCA-soluble peptides (D) of the control and *C. aurantium* juice-immersed striped catfish steaks during storage at −20 °C for 28 days.** Results are presented as mean ± standard deviation. Means in each parameter followed by different lowercase letters are significantly different ($P < 0.05$) according to DMRT.

Peroxide value (PV) is a major chemical method which indicates oxidative rancidity in fish as shown in Fig. 3B. The PV of the control and TM increased after 7 days and then decreased up to 21 days ($P < 0.05$). In addition, on days 14 and 28, no significant difference ($P > 0.05$) was found among the treatments. Our findings are in general agreement with the article of *Li et al. (2020a)* who found that the PV of Blunt snout bream (*Megalobrama amblycephala*) increased before the 8th day and subsequently decreased, without significant differences. This might have been due to the degradation of ketones, alcohols, aldehyde, and peroxides producing off-flavors in fish product.

TVB-N is related to a change in microbiological and biochemical activities (*Kyrana, Lougovois & Valsamis, 1997*). In fact, 20 mg N/100 g has been used as a point of rejection limit for fish products (*Sikorski, Kołakowska & Burt, 2020*). Considering our results (Fig. 3C), surprisingly, the TVB-N of all treatments were relevant to standard quality of fishes. The TVB-N of both treatments increased slightly on days 21 and 28, while there were significant differences ($P < 0.05$) at the end of storage (day 28) when compared to the initial date (0 day). An increase in TVB-N corresponded to a change in TCA-soluble peptides (Fig. 3D). A sharp increase in TCA-soluble peptides was found after 14 days, which indicated protein degradation. This activity was catalyzed by endogenous cathepsin during storage and by microbial proteolytic enzymes (*Xu et al., 2015*), which was related to

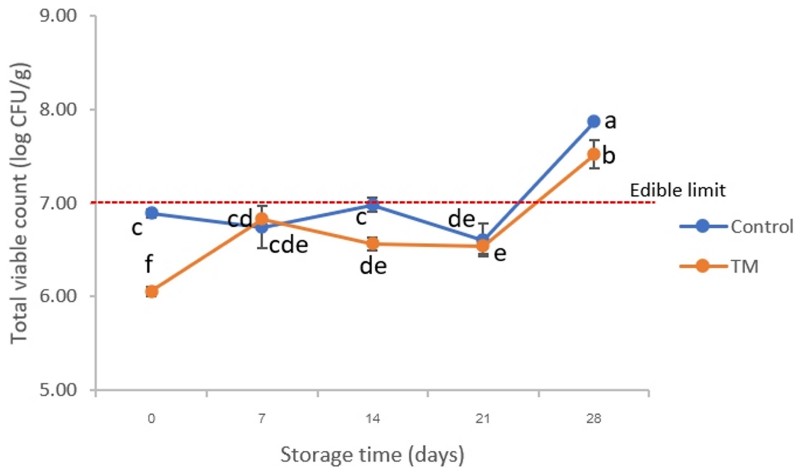

**Figure 4 Changes in total viable count (TVC) of the control and *C. aurantium* juice-immersed striped catfish steaks during storage at −20 °C for 28 days.** Results are presented as mean ± standard deviation. Means in TVC followed by different lowercase letters are significantly different ($P < 0.05$) according to DMRT.                            

the growth of total bacterial count (Fig. 4). However, it significantly decreased in both treatments on day 28 ($P < 0.05$). In the case of days 7–28, a lower accumulation of TCA-soluble peptides was shown in striped catfish steak immersed in *C. aurantium* juice. This might be interpreted that *C. aurantium* juice had the potential to retard the activity of endogenous cathepsin, and could inhibit protein-degradation microorganisms, which will be further discussed in the section of Illumina-MiSeq high throughput sequencing. In addition, major phenolic compounds of *C. aurantium* juice were found including flavonoids, *p*-Coumaric, ferulic acids, *etc.*, (*Marzouk, 2013*). *Jongberg et al. (2011)* and *Tang et al. (2016)* indicated that quinones oxidized from phenolic compounds could react with cysteine in proteins, including myofibrillar protein, which weakened the protein degradation. This also paves the strategy for industrial fish processing and preservation.

## Microbial enumeration

The results of TVC are displayed in Fig. 4. On day 0, the TVC of the control treatment and TM were approximately 6.89 ± 0.04 and 6.06 ± 0.05 log CFU/g, respectively. There were significant differences ($P < 0.05$) in the initial pH values among treatments, which correlated to the changes in pH (Fig. 3A). This indicated that the *C. aurantium* juice immersion could decrease microbiota in striped catfish steaks due to its acidic stress condition. On day 14, the TVC of the control was higher than those of TM ($P < 0.05$) and reached 6.98 ± 0.07 log CFU/g. Finally, the TVC of both treatments increased to >7.0 log CFU/g on day 28; which did not meet the edible limit of standard for freshwater fish (*ICMSF, 1986*). This phenomenon could indicate that the mesophilic bacteria were found in the late period of spoilage process on the striped catfish steaks sample under cold stress, which was consistent with the investigation of *Gram & Huss (1996)* who indicated the mesophilic microorganisms are dominant on tropical fish. This work agreed with the increasing trend in mesophilic bacteria during freezing storage which has also been

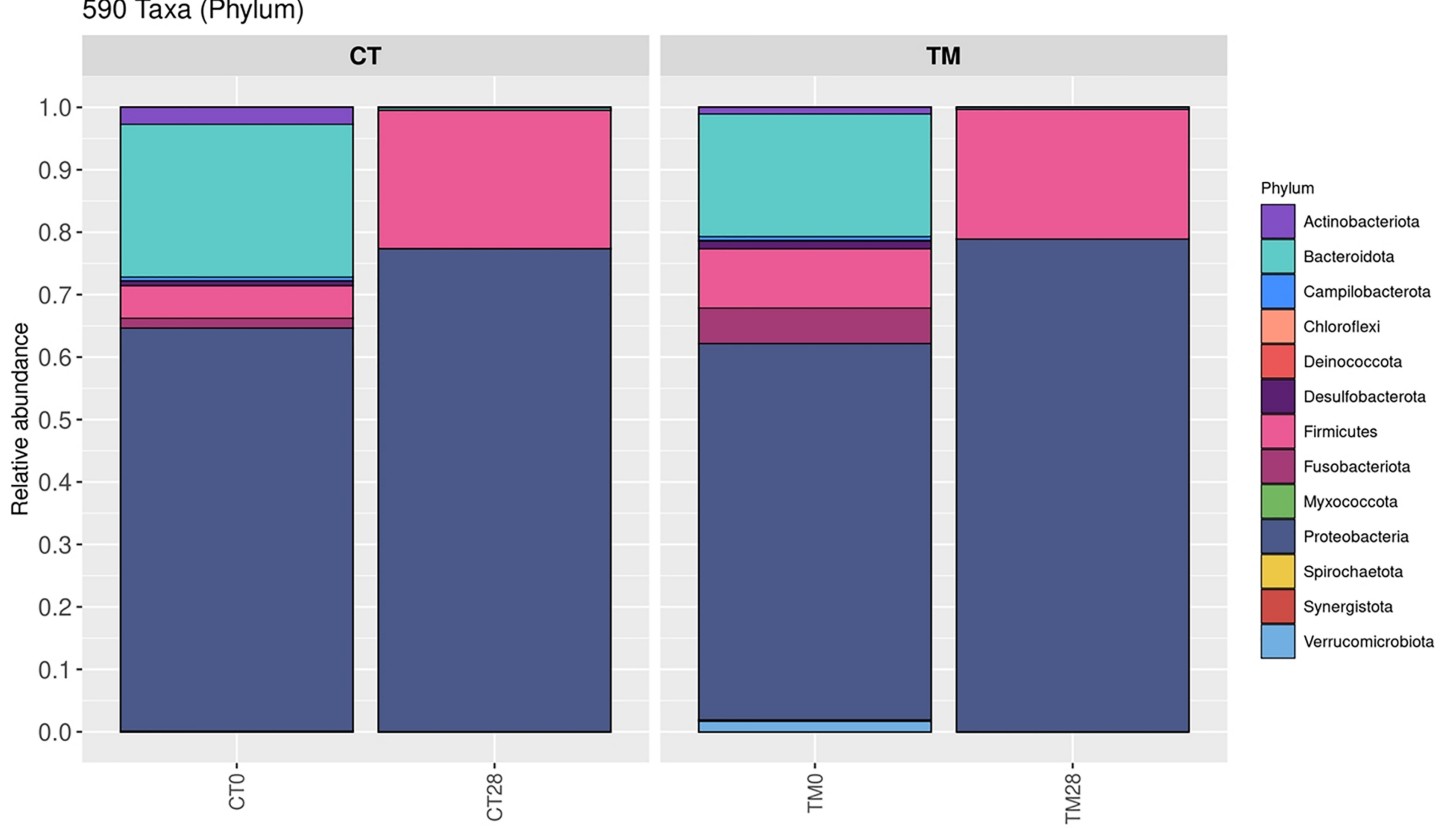

**Figure 5** Relative abundance of bacterial composition of each sample at phylum level. (CT0, control: striped catfish steak immersed in 50 ppm sodium hypochlorite on day 0; CT28, control: striped catfish steak immersed in 50 ppm sodium hypochlorite on day 28; TM0: striped catfish steak immersed in *C. aurantium* juice on day 0; TM28 striped catfish steak immersed in *C. aurantium* juice on day 28.)

observed in previous studies. It has been reported by *Ehsani & Jasour (2014)* that total viable count of silver carp (*Hypophthalmichthys molitrix*) increased after 30 days at −24 °C which did not reach the critical maximum levels of food standards (<7.0 log CFU/g). However, their findings lead us to believe that the striped catfish steaks immersed in *C. aurantium* juice should be stored at <−20 °C to prolong the shelf-life during storage.

## Bacterial community

To observe the scientific insight of the preservative effects of *C. aurantium* juice immersion, the control and TM were chosen at days 0 and 28, and their bacterial community was then identified by Illumina-MiSeq high throughput sequencing. The relative abundance of different phylum is shown in Fig. 5. Five hundred and ninety taxa at the phylum level obtained in this study were demonstrated. *Proteobacteria* (>60%) was a predominant microbiota in all samples. On day 0, *Proteobacteria* (64.55%), *Bacteroidota* (24.49%), *Firmicutes* (5.20%), *Actinobacteriota* (2.71%), and *Fusobacteriota* (1.59%) were the top five phyla of the control, while the dominant phyla responsible for top four phyla microbiota in TM were *Proteobacteria* (60.26%), *Bacteroidota* (19.65%), *Firmicutes* (9.50%), and *Fusobacteriota* (5.68%). At the end of storage time (day 28),

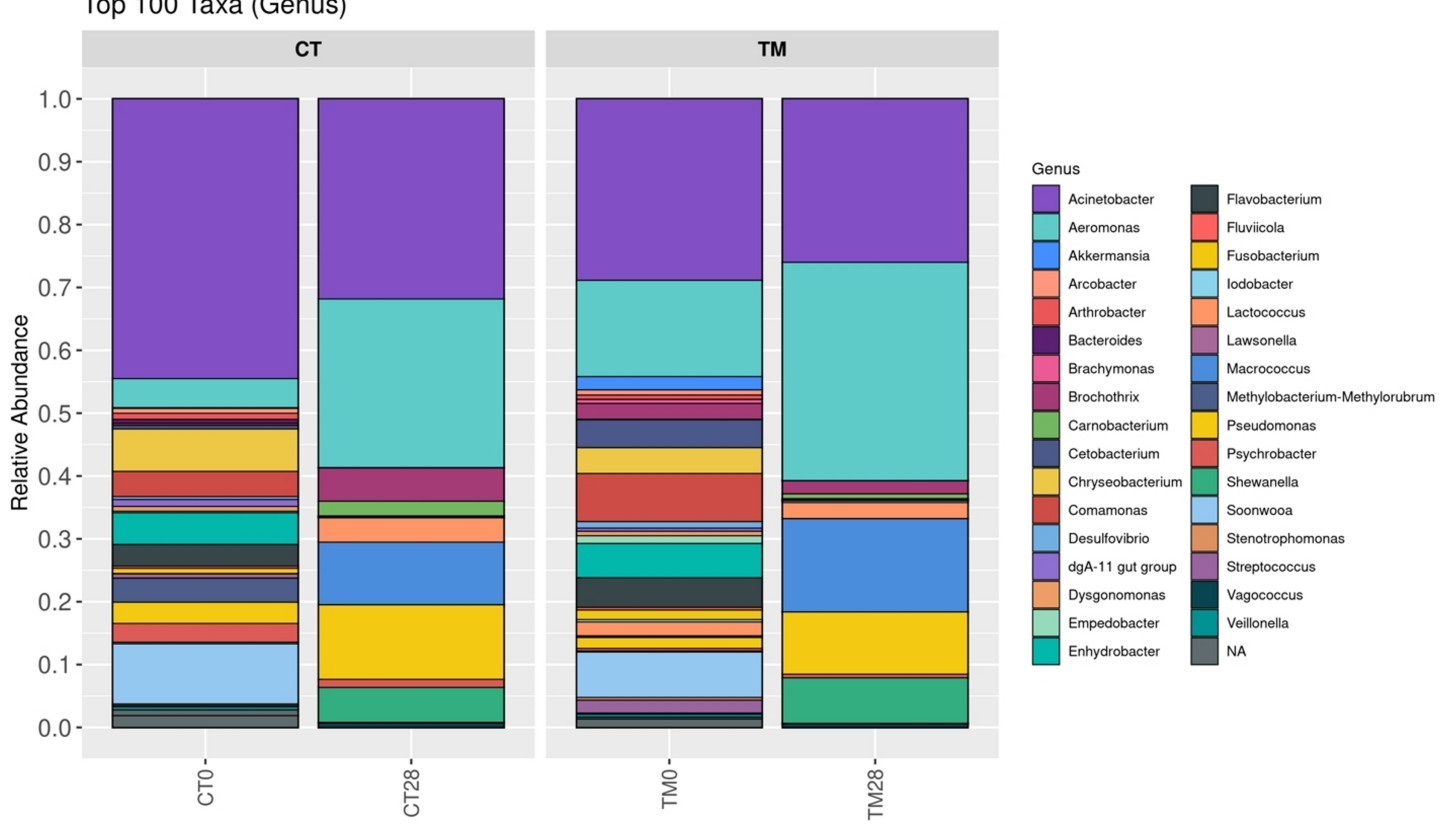

**Figure 6 Relative abundance of bacterial composition of each sample at genus level.** (CT0, control: striped catfish steak immersed in 50 ppm sodium hypochlorite on day 0; CT28, control: striped catfish steak immersed in 50 ppm sodium hypochlorite on day 28; TM0: striped catfish steak immersed in *C. aurantium* juice on day 0; TM28 striped catfish steak immersed in *C. aurantium* juice on day 28.)

*Firmicutes* (22.16%), *Bacteroidota* (0.37%), and *Actinobacteriota* (0.08%) were much more abundant in the control than in the TM treatment. In addition, a relative abundance (0.01%) of *Verrucomicrobiota, Myxococcota, Deinococcota*, and *Chloroflexi* was detected in the control treatment, but not in TM. Fig. 6 displays the relative abundance of different genera in each treatment. One hundred taxa at the genera level were specifically identified. On day 0, *Acinetobacter* (39.63%), *Soonwooa* (8.47%), *Chryseobacterium* (6.62%), *Enhydrobacter* (4.16%), *Aeromonas* (4.15%), *Flavobacterium* (4.01%), *Comamonas* (3.43%), *Pseudomonas* (3.36%), *Psychrobacter* (3.24%), and *Methylobacterium-Methylorubrum* (3.15%) were the top 10 dominant genera in the control treatment, while the TM treatment had decreased relative abundance of *Acinetobacter, Soonwooa, Chryseobacterium, Pseudomonas, Psychrobacter*, and *Methylobacterium-Methylorubrum*. However, *Acinetobacter, Aeromonas*, and *Comamonas* became the top three dominant genera in TM at day 0, which accounted for 26.27%, 12.28%, and 6.56%, respectively. On day 28, *Acinetobacter, Aeromonas, Pseudomonas, Macrococcus, Shewanella, Brochothrix, Lactococcus, Carnobacterium, Psychrobacter*, and *Vagococcus* were the top 10 dominant genera in all samples. The dominant phyla and genera of freshwater fish reported in this study were consistent with those reported by *Gonzalez et al. (2000)*,

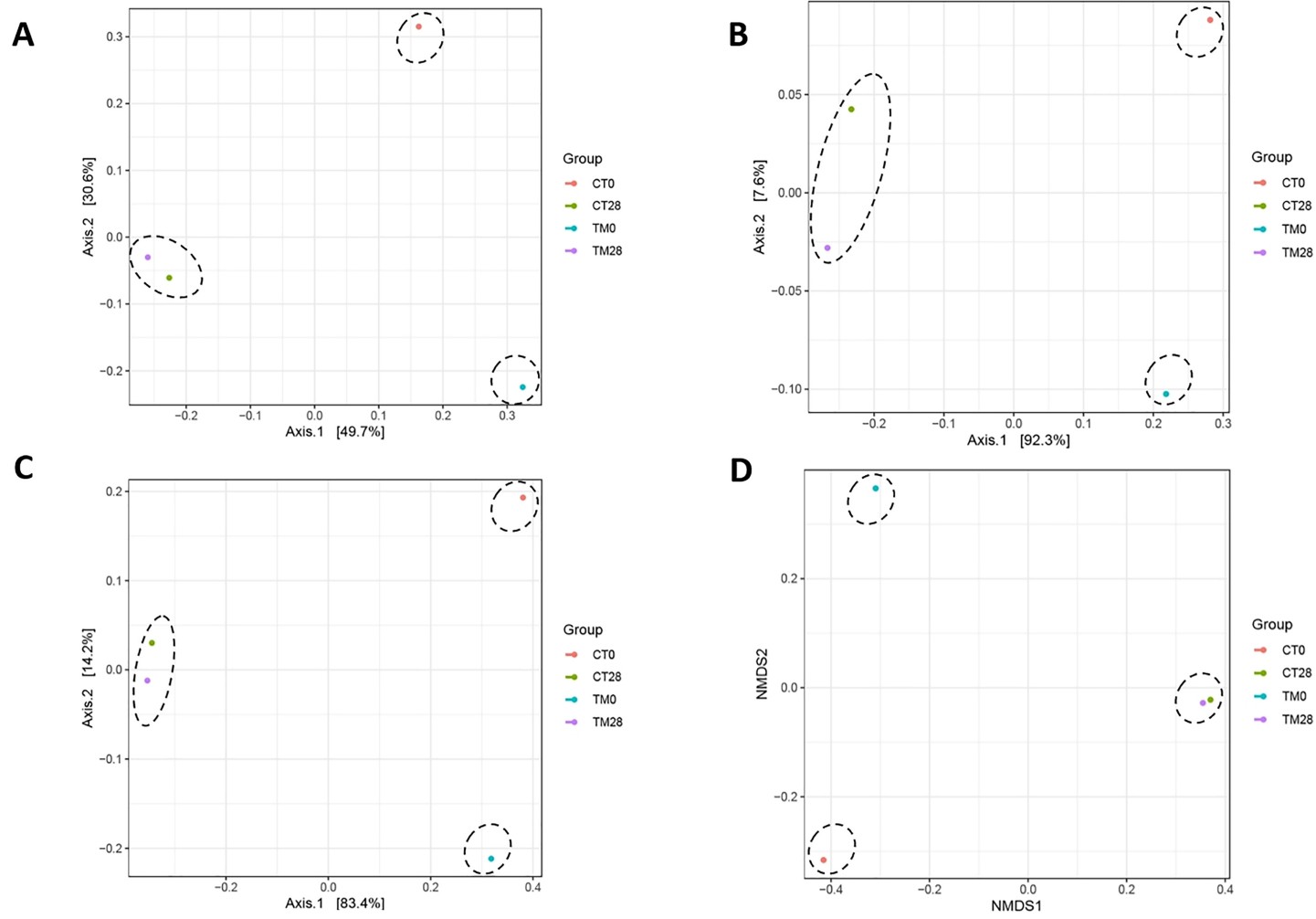

**Figure 7 Principle coordinate analysis (PCoA) plots based on unweighted UniFrac (A), weighted UniFrac (B), GUniFrac with alpha 0.5 (C), and Bray-Curtis distance (D), showing clustering of the bacterial communities from each treatment.** (CT0: striped catfish steak immersed in 50 ppm sodium hypochlorite on day 0; CT28: striped catfish steak immersed in 50 ppm sodium hypochlorite on day 28; TM0: striped catfish steak immersed in *C. aurantium* juice on day 0; TM28 striped catfish steak immersed in *C. aurantium* juice on day 28.)

*Sousa & Silva-Souza (2001)*, *Jia et al. (2018)*, and *Silbande et al. (2018)*. *Aeromonas*, *Pseudomonas*, *Lactococcus* were the major microbiota found in *Cyprinus carpio* during storage at 4 °C and −20 °C (*Li, Zhang & Luo, 2018*). Likewise, *Zhang et al. (2015)* revealed that the indigenous microbiota of fresh carp fillets had *Acinetobacter* as the major microbiota, representing 52.8% of the total isolates, while *Aeromonas* were the second most common microbiota, accounting for 21.7% of the total isolates. The *Brochothrix*, *Carnobacterium*, *Pseudomonas*, *Shewanella*, lactic acid bacteria were also identified as the dominant spoilage flora in stored striped catfish fillets (*Noseda et al., 2012*). Surprisingly, a decrease in relative abundance of *Acinetobacter, Pseudomonas, Brochothrix, Lactococcus, Carnobacterium, Psychrobacter*, and *Vagococcus* was found in TM sample on day 28 accounting for 25.48%, 9.78%, 1.98%, 2.55%, 0.73%, 0.65%, and 0.62%, respectively, when compared to the control. This phenomenon was directly related to the decrease in the

accumulation of TCA-soluble peptides (Fig. 3D) in TM owing to a decrease in the spoilage microorganisms. Furthermore, *C. aurantium* juice treatment could inhibit the growth of dominant spoilage bacteria, resulting in the different levels of relative abundance in the two treatments. According to a previous study, *C. aurantium* juice exhibited microbial activity because of the potential effect of bioactive compounds such as flavonoid and phenolic compounds. Generally, the antimicrobial properties of phenolic compounds have been revealed. The mechanism of action induces the alteration of cytoplasmic membrane, the inhibition of ion transportation and enzyme activity and then causes bacterial cell damage (*Haraoui et al., 2020*).

Figures 7A–7D present the PCoA plots that originated from unweighted UniFrac, weighted UniFrac, GUniFrac with alpha 0.5, and Bray-Curtis distance, respectively. These demonstrated the microbial community differences among the samples. On day 0, the composition of microbiome was clearly distinct between control and TM, suggesting that microbiomes among treatments were different. This might have been due to the fact that the pH condition and bioactive compounds of *C. aurantium* juice affected the microbial community of striped catfish steaks, resulting in different microbial community in each treatment. On day 28 of TM, although the relative abundance of microbial spoilages including *Acinetobacter, Pseudomonas, Brochothrix, Lactococcus, Carnobacterium, Psychrobacter*, and *Vagococcus* decreased compared to those of control (Fig. 6), the microbiota compositions of both treatments were clustered in the same group (Figs 7A–7D). This reason could imply that the application of *C. aurantium* juice as a disinfectant could replace the use of sodium hypochlorite.

## CONCLUSIONS

The preservative effect of each disinfectant agent including sodium hypochlorite (commercial disinfectant) and *C. aurantium* juice on microbial community and physico-chemical quality were revealed. Striped catfish steaks immersed in *C. aurantium* juice showed a lower accumulation of TCA-soluble peptides during storage. Moreover, *C. aurantium* juice affected the relative abundance of *Acinetobacter, Pseudomonas, Brochothrix, Lactococcus, Carnobacterium, Psychrobacter*, and *Vagococcus* on day 28. Based on the new findings, this study successfully presented *C. aurantium* juice as a feasible alternative disinfectant option for use in the preparation stage of striped catfish steaks prior to storage at $-20\ °C$.

Further studies are recommended to consider the preservative effect of specific compounds in *C. aurantium* juice on the sensorial evaluation. The specific mechanism of *C. aurantium* juice on fungi should be elucidated. Meanwhile, the effect of combining *C. aurantium* juice with other natural citrus family should be tested in striped catfish steaks to improve the color values and sensory characteristics.

## ACKNOWLEDGEMENTS

We would like to thank Miss Martha Maloi Eromine for English language editing of this manuscript.

### Funding

This work was supported by Rajamangala University of Technology Isan, Nakhon Ratchasima, Thailand. The funders had no role in study design, data collection and analysis, decision to publish, or preparation of the manuscript.

### Grant Disclosures

The following grant information was disclosed by the authors:
Rajamangala University of Technology Isan, Nakhon Ratchasima, Thailand.

### Competing Interests

The authors declare that they have no competing interests.

### Author Contributions

- Kajonsak Dabsantai performed the experiments, analyzed the data, authored or reviewed drafts of the article, and approved the final draft.
- Thitikorn Mahidsanan conceived and designed the experiments, analyzed the data, prepared figures and/or tables, authored or reviewed drafts of the article, and approved the final draft.

### DNA Deposition

The following information was supplied regarding the deposition of DNA sequences:
The sequences are available at the National Center of Biotechnology Information (NCBI) Sequence Read Archive (SRA): PRJNA914840.

### Data Availability

The raw data are available in the Supplemental Files.

### Supplemental Information

Supplemental information for this article can be found online at http://dx.doi.org/10.7717/peerj.15168#supplemental-information.

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
