# Peer review of "Effect of Citrus aurantium juice as a disinfecting agent on quality and bacterial communities of striped catfish steaks stored at −20 °C"

_PeerJ, doi:10.7717/peerj.15168_

## Round 0.1 · original submission · Major Revisions

Dear Authors Thank you for your recent submission to PeerJ. Your manuscript has been examined by expert reviewers whose reviews are attached. Based on the reviewer reports and my own reading of the manuscript, we find that major revisions are required for your manuscript to be suitable for publication.
Kindly revise your manuscript and modify it according to the attached comments, besides providing a point-by-point response in a separate file along with the revised version of the manuscript.
Best regards

·

Basic reporting

This manuscript reports original research findings aimed at testing the hypothesis that bitter orange juice could be used to disinfect striped catfish prior to frozen storage for up to 4 weeks. Although it tackles a relevant problem using a scientifically sound methodology, this text needs several improvements and completions, as explained below:

1. To structure this manuscript according to PeerJ's standards, please report the "Results" separately from "Discussion". This will help the reader to assess the results based on her/his own background.

2. To make the statistical analysis transparent and reproducible, please also provide the ANOVA tables and DMRT output as Supplementary Materials.

3. To comply with PeerJ's open data policy, please provide the raw data in a clearly organized Excel spreadsheet.

4. Please explain all acronyms used in the text. For example, on line 147, instead of "TCA-soluble", please write "trichloroacetic acid (TCA)-soluble". Then, on line 149 "trichloroacetic acid" should be replaced by "TCA". Also, several explanations are needed for the acronyms used in the description of Figure 7.

5. The opening sentence of subsection 2.3.2 (line 160) needs revision.

6. The sentence from line 258 is vague and lacks a reference.

Experimental design

The Introduction is well written and clearly identifies the gap of knowledge addressed by the study. The Materials and Methods section, however, is too laconic.

1. Please provide a brief description of the method used to assess the amount of total volatile basic nitrogen. The steam distillation assay, proposed in 1968 by the Codex Alimentarius Committee and used in (Malle and Poumeyrol, 1989), involves a specific equipment, which needs to be specified for the sake of reproducibility.

2. The Lowry method, mentioned on line 150, should be described in more detail.

3. Section 2.4 needs to be supplemented with a brief description of the Principal Coordinate Analysis (PCoA).

Validity of the findings

1. In "Discussion", please comment on the oscillatory pattern displayed by the control group in two color space coordinates (a* and b*). It is puzzling that statistically significant differences were observed between the colors of control and C. aurantium juice-treated stripes after 2 and 4 weeks, but not after 3 weeks. Do the other techniques suggest an explanation of this unsteady time course of the color of specimens treated with sodium hypochlorite?

2. The last section ("Conclusions") is loosely connected with the rest of the manuscript. In a single sentence, it deems bitter orange juice a "feasible alternative" to sodium hypochlorite. For more specific conclusions, similarities, differences, advantages and disadvantages of the new solution need to be addressed in light of the reported results.

Reviewer 2 ·

Basic reporting

1. Clear and unambiguous English language is used throughout the article.

2. This article provides sufficient evidence to suggest that C. aurantium has the potential to replace the chemical preservatives.

3. Line 72: …..regarding safety concerns for food & health…..
Please elaborate more on what are the safety concerns for health & food. Explain with respect to the effect of excess chlorine on fish products during preservation, release of chlorine/ chlorinated water in the environment and its impact on aquatic flora and fauna etc.

4. Line 74: “To overcome this limitation…..”. I recommend replacing the word, limitation with ‘hazards of the chemical preservative’.
5. Line 225-226: Please provide a reference why European Union is concerned of safety aspect of sodium hypochlorite.
6. Based on your findings, will the concentrated C. aurantium juice be a better preservative? Also comment on prospects of usage of combinations of natural products like C. aurantium & lemon juice etc. Example: PMID: 26396373
7. Did you detect TBARS (Thiobarbituric Acid Reactive Substances) -which is an indicator of lipid oxidations?

Experimental design

1. The experimental design is orderly, with stepwise relevant experiments.
2. Line 125: Why 28 days were chosen for the end-point analysis? Please clarify if catfish can be frozen only for that period of time.
3. Line 128 & 132: I appreciate that color & texture analyses were not done manually, and dedicated instruments were used. This reduces the personal bias and is a robust and reliable way of data analysis.
4. It is not clear if all the data analysis was done blindfolded.
5. Line 136: Please elaborate more on AOAC and give appropriate reference.
6. Line 153: For microbial enumeration, do you think anaerobic bacteria in the fish gut might also play a role in food spoilage? Also, I would recommend mentioning about slow-growing organisms like fungi, where 24 hours incubation might not be sufficient.
7. Why only 3 fish steaks from each group were chosen for the experimental analysis? And were the 3 fish steaks chosen randomly?

Validity of the findings

1. Robust methods are used to analyze the Physico-chemical properties of the fish steaks.
2. Data analysis is performed using Statistical measures.
3. Though I can access the raw sequence reads on NCBI, I would recommend a mention of the same (Bioproject Number and SAMNXXXX links) etc. in the main text.

Additional comments

Nothing to add.

---

## Round 0.2 · Minor Revisions

Dear Dr. Mahidsanan,

Thank you for your submission to PeerJ.

It is my opinion as the Academic Editor for your article - Effect of Citrus aurantium juice as a disinfecting agent on quality and bacterial communities of striped catfish steaks stored at -20 °C - that it requires a number of Minor Revisions.

·

Basic reporting

Except for separating "Results" from "Discussion", the authors implemented all the recommendations from my previous referee report. Although, as a reader, I still prefer them separated, I see no technical reasons to reject this article.

Experimental design

The Materials and Methods section has been extended taking into account all my concerns.

Validity of the findings

The authors did a good job revising and supplementing the discussion and conclusions. Also, the raw data and statistical computations are presented in the supplemental information. I have no further comments regarding the validity of the reported results.

Additional comments

I think this is a thorough study of a well-defined problem.

Reviewer 2 ·

Basic reporting

1. Line 28: It is not clear if you’re talking about TM- 14 days or 28 days. Be precise.
2. Line 30: I would recommend to comment on peroxide values of TM – 14 days and 28 days relative to control.
3. Line 31: It is confusing about the “TM” mentioned for lower levels of tri-chloro acetic acid soluble peptides.
4. Line 38: If total viable count exceeded the edible limit (as mentioned in Line 34), how would the fruit juice replace the chemical preservative.
5. Line 72: “ A recent study…….”. The reference provided is of 1973, which can’t be stated as the ‘recent study’.

Experimental design

1. Line 129: Were the steak samples chosen randomly and in a blindfolded manner to analyze the phyisco-chemical/ microbiologcal properties?

Validity of the findings

No comments.

Additional comments

None.

---

## Round 0.3 · accepted · Accept

Dear Dr. Mahidsanan,
Your manuscript - Effect of Citrus aurantium juice as a disinfecting agent on quality and bacterial communities of striped catfish steaks stored at -20 °C - has been Accepted for publication. Congratulations!